# Isoline Tracking in Particle-Based Fluids Using Level-Set Learning Representation

**Jun Yeong Kim [1], Chang Geun Song [1], Jung Lee [2], Jong-Hyun Kim [3], Jong Wan Lee [4] and Sun-Jeong Kim [1,*]**

[1] Department of Convergence Software, Hallym University, Hallimdaehak-gil, Chuncheon-si 24252, Republic of Korea; wnsdud5645@gmail.com (J.Y.K.); cgsong@hallym.ac.kr (C.G.S.)

[2] Department of Computer Science, Hanbat National University, Dongseo-daero, Yuseong-gu, Daejeon 34158, Republic of Korea; airjung@hanbat.ac.kr

[3] College of Software and Convergence (Department of Design Technology), Inha University, 100, Inha-ro, Nam-gu, Incheon 22212, Republic of Korea; jonghyunkim@inha.ac.kr

[4] School of Nano Convergence Technology, Hallym University, Hallimdaehak-gil, Chuncheon-si 24252, Republic of Korea; jwlee@hallym.ac.kr

[*] Correspondence: sunkim@hallym.ac.kr

**Abstract:** In this paper, we propose a learning model for tracking the isolines of fluid based on the physical properties of particles in particle-based fluid simulations. Our method involves analyzing which weights, closely related to surface tracking among the various physical properties of fluid particles, are significant. These weights are used as input values for the learning algorithm, enabling relatively accurate isoline tracking. In addition, compared to existing learning models such as linear regression, LSTM (long short-term memory), and learning representation (1-layer) models, our method obtained superior surface tracking results without accumulating errors. By using our proposed network structure to track the fluid surface, it learns and predicts values derived from existing fluid simulation algorithms, eliminating the need for computational processes for level-set values and enabling real-time surface tracking. As the scale of the simulation increases, our method significantly reduces the time and resources consumed compared to traditional methods and can track the fluid surface without additional resource consumption. Furthermore, due to our method's simple network structure, the time consumed in the initial process of loading the model into memory is faster than models such as CNN and LSTM. Our proposed model occupies less than 30 kb of storage space, making it suitable for use in middleware. Lastly, to verify the generality of our method, we conducted tests in a total of five scenes, and in all test scenes, visually natural fluid isolines were represented.

**Keywords:** isoline tracking; fluid simulation; fluid surfaces; level set; artificial neural network; particle-based fluids

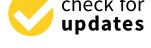



## 1. Introduction

A realistic and stable real-time representation of fluid has been a long-standing research issue in the field of Computer Graphics [1–4]. MC (Marching Cubes), a representative method for representing the surface of fluids, has the disadvantage of losing the model's sharp features [5]. DC (Dual Contouring), which overcomes this drawback, can accurately represent sharp features of objects using QEF (Quadratic Error Function), but surface normalization must precede to calculate the appropriate vertex positions [6]. Consequently, approaches that implicitly generate surfaces need to find suitable data structures to identify more accurate vertex positions and solve optimization functions, and this process requires increasing resources and computation time as the number of polygons increases.

Fluids simulated using the SPH (Smoothed Particle Hydrodynamics) technique, unlike typical 3D models with point cloud structures, require visually representing only the surface of the fluid's point cloud, excluding the interior [7,8]. This necessitates an additional process

of extracting fluid surfaces. The reconstruction of fluid surfaces can only be done after calculations involving the fluid particles' position, density, viscosity, pressure, surface tension, and external forces are completed. The time and resources needed for simulation and surface extraction depend on the dimension of the space and the number of particles [9]. Unlike standard 3D models, which usually remain static over time, fluids continuously change their surfaces and interiors due to particle movement, requiring constant tracking of the surface in each frame. Consequently, these processes demand significant resources and computation time. Although advancements in hardware and CG technology have made realistic graphics widely accessible, mobile devices like smartphones, with limited resources, struggle to represent high-quality fluids in real time. The increased computation time due to limited resources forces a reduction in content quality to maintain real-time continuity in rendering. This quality reduction can decrease user engagement and make continuous use of the content challenging. Therefore, from a commercial perspective, it is essential to maximize graphics quality with limited resources to ensure sustained user engagement with the content.

In this paper, we propose a learning model based on the physical properties of fluid particles obtained from SPH (Smoothed Particle Hydrodynamics)-based fluid simulations for tracking the surface of these particles. The proposed model reconstructs surfaces in real time with quality similar to ground truth in middleware environments. It learns physical properties such as position, density, the number of neighboring particles, and pressure used in SPH-based fluid simulations during preprocessing, thus avoiding additional resource consumption for surface tracking during runtime. Our method, by inferring level-set values used in surface tracking through the trained model, eliminates the level-set calculation process included in traditional methods, enabling real-time surface tracking of fluids in SPH-based simulations with fewer resources.

To explore if similar quality can be achieved with fewer resources and a simpler structure than our proposed method, we implemented linear regression and learning models. We also integrated an LSTM (Long Short-Term Memory) model into the surface tracking structure of SPH-based fluid simulations to see if learning and predicting time-dependent characteristics of fluid could improve surface tracking accuracy. To prevent unnecessary learning in our model, we used a heat map to extract correlations among features used in input, thus identifying feature trends for surface tracking predictions. We also examined the error changes after sequentially removing and retraining each input feature, to identify which ones most significantly influence learning.

To evaluate the generalizability of our proposed model, we conducted tests in five different scenes, excluding the scenes used for training. By eliminating the level-set calculation process included in traditional surface reconstruction techniques, we were able to track and reconstruct the surface of fluids in real time using fewer resources. Consequently, we visually confirmed that our method could reconstruct fluid isolines of similar quality to those found in existing research.

## 2. Related Work

### 2.1. Surface Reconstruction of Particle-Based Fluids

Originally developed to solve problems in astrophysics, the SPH technique [7], a prominent particle-based fluid simulation method, became popular in fluid simulations due to its ability to assign physical properties to particles instead of a mesh. This led to research into various methods of solving partial differential equations using SPH [10–13]. The instability of fluid simulations and their significant resource consumption make the search for methods to achieve stable simulations within domains [14–17] and natural representation of particle-based fluids at low cost [18–20] an ongoing popular topic. In typical fluid simulations, techniques are used to realistically represent the fluid surfaces based on the material properties of certain particles or to reflect different characteristics like honey or water in the physical properties affecting particle movement [2,13,21]. Extensive research focuses on reconstructing meshes from point cloud structures to reduce resource

usage and computation time while visually representing fluids naturally. These studies are widely referenced in related research fields.

To represent the surface of a fluid, it is necessary first to identify surface particles among all fluid particles. In grid-based simulation frameworks, methods like the level-set method [22], particle level set [23–25], and VOF (Volume of Fluid) [26] have been proposed. In the meshless (Lagrangian) simulation framework used in this paper, Blinn [27] introduced the blobby sphere approach. This method extracts surfaces from a scalar field composed of the sum of radial functions centered on each particle. However, it faced issues of surface irregularities when particle density was high or low. Zhu and Bridson [28] obtained smoother surfaces by computing scalar values from basis functions, which are the weighted sum of neighboring particles' weights and radii, and then sampling this scalar distance function on a grid followed by a smoothing pass. Later, Adams [29] improved this method proposed by Zhu and Bridson by tracking the particle-surface distance every frame. This approach adjusted the distance of each particle to the surface and propagated the distance information from surface particles to internal particles, thereby realigning the particles. This method also enables smooth surface generation for both fixed-radius particles and adaptive particle size in fluid simulation techniques [20] that increase memory efficiency and reduce computational costs.

MC [5] was proposed to efficiently construct high-resolution 3D surfaces. When using MC to generate surfaces, the volume of an object is divided into small cubic voxels. The algorithm constructs triangles based on a pre-allocated MC table in memory, approximating the surface within each voxel. These triangles are created by interpolating vertices on the edges of voxels based on the data values at their endpoints, and connecting the results in a smooth, continuous surface across the entire volume. While individual voxel computations allow for easy parallel processing, complex topologies may lose features like sharp edges. Also, additional information for each voxel, including vertex position, normals, and connectivity data, must be stored, increasing the required memory amount for high-resolution grids and large-scale data.

DC [6] addresses the loss of sharp edge features in MC by effectively handling complex topologies through the integration of data value gradients. It does not solely rely on predefined tables but iteratively evaluates data points within each voxel, calculates gradients, and determines intersections where the surface crosses voxel edges. Utilizing these calculated intersections can enhance accuracy, but it also increases computational complexity. Efficient implementation requires additional optimization techniques, and there are some constraints to consider.

### 2.2. Fluid Simulation with Deep Learning

With the significant advancements in various fields using deep learning, research has been conducted in fluid simulation to learn physical properties through it, aiming to gain benefits in memory efficiency and simulation acceleration [30]. To accelerate grid-based fluid simulations, the CFDNet (Computational Fluid Dynamics Network) [31], utilizing a CNN (Convolutional Neural Network), was proposed. This model inputs the initial and boundary conditions of fluid simulations, infers the fluid's dynamic movements, and achieves fast and accurate results using deep learning, outperforming traditional numerical analysis methods. Experimental results showed that CFDNet performs fluid simulations significantly faster than existing methods. This approach aimed to speed up simulations and minimize errors in the model, and demonstrated the potential of learning the physical properties of fluids through CNN.

In Lagrangian fluid simulations, FluidMLP [32] was proposed to generate a model by learning particles' physical properties. This model, which learned and compared particles' positions, velocities, and accelerations, improved accuracy notably when using acceleration data. However, it faced accuracy issues during particle interactions, causing particles to escape the simulation boundaries. To resolve this, the FGN (Fluid Graph Networks) model [33] was designed with an Edge-focused network for learning particle interactions

and a Node-focused network for learning particles' physical properties. The model, created using GNN (Graph Neural Network), demonstrated error convergence within a certain range across various environments and confirmed its capability to learn physical properties of fluid simulations. FluidMLP and FGN, by leveraging models generated through deep learning, significantly reduced simulation computational time by bypassing traditional fluid simulation algorithms. However, due to the importance of accuracy in simulations, extensive research is ongoing to achieve a level of accuracy comparable to that of existing fluid simulation algorithms.

Research applying deep learning to enhance memory efficiency and reduce computational time has been conducted for the surface reconstruction process in fluid simulations. To overcome the issue of losing sharp edge features in results from traditional MC, NMC (Neural Marching Cubes) [34] was proposed, utilizing mesh vertex positions and topology as training data. Although NMC provides more accurate surfaces than MC, it uses more complex tessellation templates than MC, leading to the generation of more triangles and requiring over 100 times more computational time for mesh reconstruction. Additionally, this method assumes the implicit function and signed distance field are provided together, so it only learns aspects related to the MC process.

NDC (Neural Dual Contouring) [35] is a data-centric model based on DC. A common drawback of MC and DC is the need for surface normalization to calculate the optimal vertex position in each cell. However, NDC, as a data-centric model, bypasses this process and predicts vertex positions without it. The model, trained through CNN, results in 3 to 20 times faster inference times compared to NMC and generates 4 to 8 times fewer meshes, thus increasing memory efficiency. Based on such prior research, this study aims to replace traditional surface tracking algorithms in fluid simulations with learning models, thereby reducing computation time and enhancing memory usage efficiency.

## 3. Proposed Framework

This section discusses the methods used to extract data for learning and describes the issues with surface tracking techniques used in traditional methods. It also explores the model proposed in this study.

### 3.1. A Review of Fundamentals in Particle-Based Fluids for Data Acquisition

In this paper, data are generated using SPH-based fluid particle simulation [10]. This method quickly acquires physical quantities such as density and pressure held by particles using kernels during calculations. Stability is also enhanced by designing kernels that take into account the physical properties of various forces (e.g., pressure, viscosity, surface tension). The equation below describes the basic kernel form of SPH (see Equation (1)).

$$A_S(r_i) = \sum_j m_j \frac{A_j}{\rho_j} W(r_i - r_j, h) \tag{1}$$

where $A_j$ represents the field quantity at position $r_j$, $r_i$ is the position of the target particle, $m_j$ is the mass of the neighboring particles, and $\rho_j$ is the density of the neighboring particles. Additionally, $r_j$ indicates the position of the neighboring particles, $h$ is the smoothing length, and $W$ denotes the kernel function. The range within which neighboring particles are found is determined by the size of $h$, and the influence is calculated through $W$, a distance-based weight between the target particle and the neighboring particles. As described in Equation (1), the kernel function $W$ plays a significant role in the movement of particles. Therefore, the design of $W$ can create simulations with desired characteristics. The following is the SPH kernel used in this paper (see Equations (2)–(4)).

$$W_{poly}(r, h) = \frac{315}{64\pi h^9} \begin{cases} (h^2 - r^2)^3 & 0 \leq r \leq h \\ 0 & otherwise \end{cases} \tag{2}$$

$$W_{spiky}(r,h) = \frac{15}{\pi h^6} \begin{cases} (h-r)^3 & 0 \le r \le h \\ 0 & otherwise \end{cases} \tag{3}$$

$$W_{viscosity}(r,h) = \frac{15}{2\pi h^3} \begin{cases} -\frac{r^3}{2h^3} + \frac{r^2}{h^2} + \frac{h}{2r} - 1 & 0 \le r \le h \\ 0 & otherwise \end{cases} \tag{4}$$

In the above equation, $W_{poly}$, $W_{spiky}$, and $W_{viscosity}$ are used for calculating density, pressure, and viscosity, respectively, where $r$ is $r_i - r_j$. To train the model proposed in this paper, data acquired from the simulation, such as position, density, the number of neighboring particles, and values of the state equation used in input computation, are utilized. All these values influence the change in density. The level-set values calculated through density serve as a criterion for distinguishing the fluid's free surface and represent the surface through the boundary of classified particles. Learning the changes in density over time allows the model to identify the surface. The state equation used in this paper is as in Equation (5).

$$p = k(\rho - \rho_0) \tag{5}$$

where $p$ represents pressure, $k$ is the gas constant, $\rho$ is the density, and $\rho_0$ denotes the rest density.

### 3.2. Mesh Generation with Generative Artificial Intelligence

Jin et al. presented a method that improves the accuracy and speed of surface reconstruction by interpolating the position of query points through multiple grids and learning the SDF (Signed Distance Function) values via LNN and LSL [36]. However, they trained their models using mesh-type data, and physical characteristics such as density, pressure, or the number of neighboring particles, which are used in this paper, were not utilized as learning data.

Zhou et al. introduced a new technique called Level Set Projection (LSP) utilizing Zero Level Set (ZLS) [37]. While traditional SDF methods had discontinuity issues at the Zero level set, LSP addressed these discontinuities by using the gradient of SDF and employed UDF (Unsigned Distance Functions) as learning data. This method excellently differentiates and reconstructs surfaces, but it applies only to static objects.

Additionally, Shao et al. proposed the INPM (Improved Neural Particle Method), based on the network of PINN (Physics-Informed Neural Networks) [38]. They trained physical properties to distinguish free surface particles existing on the surface. However, while their model classifies particles, its ultimate goal is only to differentiate surface particles, meaning additional steps are required to restore fluid isolines.

### 3.3. Isoline Tracking in Basic Particle-Based Fluids

The level-set method is used in various fields like Fluid Dynamics, Computer Graphics, and Computer Vision. The essence of this method is encoding fluid surfaces implicitly, rather than representing them explicitly, by defining a level-set function across the entire simulation domain. Implicitly represented fluid surfaces are suitable for handling topological changes like the merging or splitting of surfaces. This approach allows for the natural representation of complex shapes and boundaries that change over time without the need to reproduce meshes or reparameterize. In this paper, the level set was calculated using the implicit function proposed by Zhu and Birdson [28].

Surface tracking identifies and represents the boundary between the surface and the air layer. The method used for surface tracking determines the boundaries of the surface, so it is crucial to choose a method suitable for the fluid characteristics to naturally represent fluid surfaces. When tracking fluid surfaces using the methods mentioned earlier by Blinn [27] and Müller [10], noise can sometimes appear on stable fluid surfaces (see Figure 1).

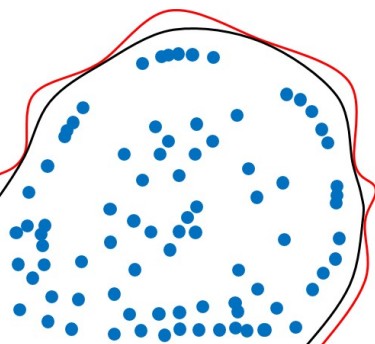

**Figure 1.** Blinn's method [27] represents the outer curve (red), while Zhu and Bridson's method [10] represents the inner curve (black).

To mitigate this issue, Zhu and Bridson [28] proposed the following level-set function (see Equation (6)).

$$\phi(x) = |x - \bar{x}| - \bar{r} \tag{6}$$

In traditional methods, $\bar{x}$ and $\bar{r}$ were the position and radius of a single particle, respectively. These were replaced with $\bar{x}$ and $\bar{r}$, the average position and radius weighted by neighboring particles, to reduce the influence of high-density areas and enable smooth surface representation. The equation was designed as follows to control the influence of particles (see Equations (7)–(9)).

$$\bar{x} = \sum_i w_i x_i \tag{7}$$

$$\bar{r} = \sum_i w_i r_i \tag{8}$$

$$w_i = \frac{k(|x - x_i|/R)}{\sum_j (|x - \bar{x}_j|/R)} \tag{9}$$

where $w_i$ represents the weight, and $k$ is defined as $k(s) = max\left(0, \left(1 - s^2\right)^3\right)$. A notable aspect of this equation is the significant computational cost involved in calculating the weights $w_i, \bar{x}, \bar{r}$. To compute $\bar{x}$ and $\bar{r}$, the values of $w_i$ must be determined first, and to know $w_i$, $k(s)$ needs to be calculated. Especially in high-resolution simulations or large scenes, the requirement for frequent re-initialization and advection steps leads to increased simulation time and memory demand. Therefore, to represent high-quality fluid surfaces in real time, the burden of resources needed to compute the level set must be reduced.

*3.4. Learning Representation for Isoline Tracking*

The proposed method aims to efficiently represent isolines in 2D by learning the level-set values needed for surface reconstruction through network training. To achieve this, the model structure is designed to be as simple as possible to reduce memory usage and ensure operability in general test environments. It should also track and reconstruct surfaces with accuracy comparable to traditional methods. To reduce wasted resources, it is essential to analyze the correlation of input data used for level-set inference to prevent unnecessary learning, and experiment with various approaches to find the optimal layer structure and parameters in the neural network for effective model training.

Figure 2 represents the network structure of the model proposed in this paper. By learning and inferring values derived from traditional algorithms when tracking the fluid surface with the proposed network structure, the computational process for level-set values is eliminated, enabling real-time surface tracking. As the scale of the simulation increases, the time and resources consumed can be significantly reduced compared to traditional approaches. The physical data used as input are processed during the preprocessing phase

of the simulation, allowing the fluid surface to be tracked during runtime without additional resource consumption. Furthermore, the simplicity of our network structure occupies less than 30 kb of storage space, making the model's initial loading into memory faster than models like CNN and LSTM, and thus suitable for use in middleware.

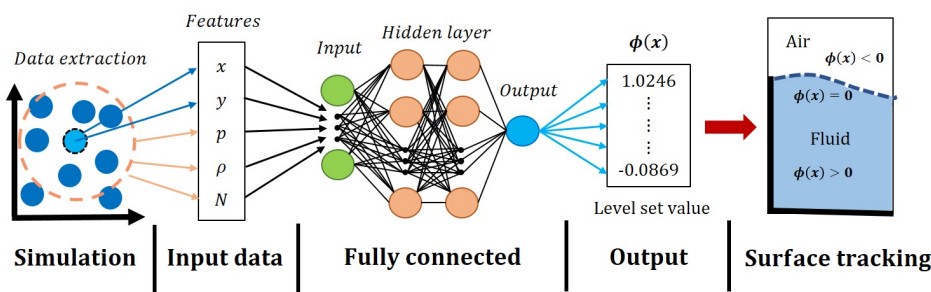

**Figure 2.** The network architecture of the proposed model. $x$ and $y$ are the positions of the particles, $p$ is the pressure, $\rho$ is the density, and $N$ is the number of neighboring particles. It runs from left to right.

Feature Analysis

For efficient learning, it is crucial to understand the data form of the level set and the variables used in its computation. Considering the characteristic of the level set, which suggests that "the more neighboring particles a target particle has, the more likely it is to be inside the fluid", it is evident that the distribution of particles is important. The distribution of particles can be represented by density, and the key idea of this paper starts from the assumption that density will significantly contribute to the inference of the level set.

As the number of layers and the input data in the learning process increase, the training time grows, and the model becomes more complex. Therefore, if the input data and the number of layers can be appropriately reduced, accuracy and learning speed can be improved, and unnecessary resource consumption can be minimized. Figure 3 displays a heat map representing the correlation of the input features. A value close to 1 or −1 for an input feature when the level-set increases indicates a positive or negative correlation, respectively. Conversely, a value close to 0 implies a nonlinear relationship with the level-set value, which could adversely affect model accuracy during training. The selection criteria for the frame to determine the correlation is a scene where the ratio of interior particles to surface particles is balanced. As the variance of particles increases along the $Y$-axis, the number of surface particles also increases. A balanced ratio between surface particles and interior particles can prevent learning biases caused by imbalanced data, aiding in the model's generalization. Hence, scenes with widely distributed particles are used. Table 1 lists the features used for training.

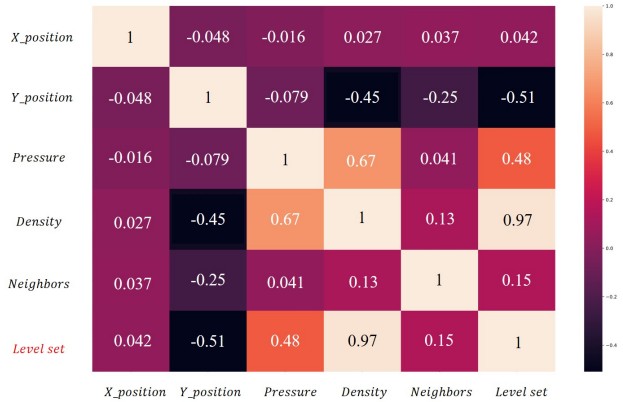

**Figure 3.** Correlation heat map of input features.

**Table 1.** The features used in training data.

| Input Feature | Symbolic Meaning |
| --- | --- |
| $x$ | X-position of particle |
| $y$ | Y-position of particle |
| $p$ | pressure |
| $\rho$ | density |
| $N$ | number of neighboring particles |

*3.5. Comparison of Learning Models*

In this section, experiments are conducted by modifying the structure of various layers and parameters to compare and identify better methods relative to the one proposed in this paper. For comparison, models such as linear regression, learning representation, and LSTM are used. Learning representation involves models with 1-layer and our method, respectively.

Linear regression and learning representation (1-layer) were designed to check if the input data structure could be represented in a lighter structure than that proposed by the model, while LSTM was selected as part of the experimental group to explore its ability to ensure temporal continuity in simulations.

Figure 4 shows a model using LSTM to infer level-set values, referencing the method of improving prediction accuracy through Warm-up in CFDNet [31]. LSTM, used for learning sequence data, requires the use of some inferred values as input due to its time-dependent characteristics. As the calculated level-set values increase, the resource consumption used for tracking the fluid surface grows; thus, the sequence data used in LSTM were limited to three frames to ensure meaningful experimental results. Following are the equations for RMSE (Root Mean Square Error) and MSE (Mean Square Error) used for evaluation (see Equations (10) and (11)).

$$RMSE = \frac{1}{N}\sum_{j}^{N}\sqrt{\left(|y_t| - |y_p|\right)^2} \tag{10}$$

$$MSE = \frac{1}{N}\sum_{j}^{N}\left(y_t - y_p\right)^2 \tag{11}$$

where $N$ represents the total number of particles, $y_t$ is the ground truth, and $y_p$ is the prediction value.

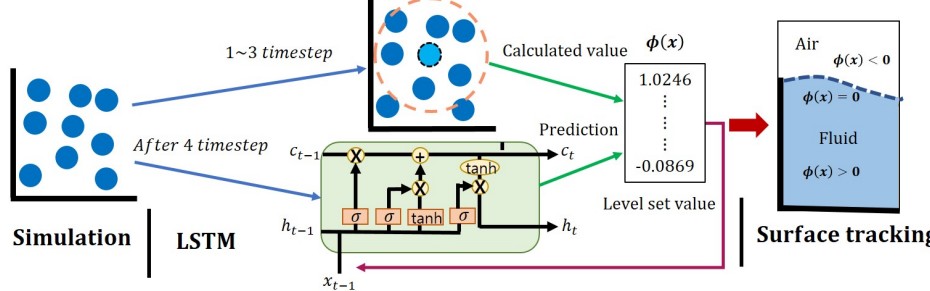

**Figure 4.** Surface tracking structure of the LSTM model (top: previous algorithm, bottom: LSTM model). It is used for prediction after calculation with previous algorithm.

## 4. Experiments

This section investigates the changes in loss when certain input values are excluded from the learning process.

Table 2 compares loss values by sequentially removing each feature used in training. The lowest loss occurred when $x$, which had the lowest correlation with the level-set value,

was removed from the input features. Although the model trained without $x$ showed the smallest loss, it was determined that the difference between validation loss and training loss was not significantly meaningful. Furthermore, it cannot be assured that excluding $x$ always results in the smallest loss across various scenes when calculating validation loss, so the paper opts to train the model using all features. The most significant change in loss occurred when density was removed from the input features. Based on these results, it was conjectured that density might be an essential element that must be included in learning for level-set inference. "None" in Table 2 refers to the results when all features were used as training data.

**Table 2.** Training loss and validation loss after removing a specific feature.

| Excluded Feature | $x$ | $y$ | $p$ | $\rho$ | $N$ | None |
|---|---|---|---|---|---|---|
| *Training loss* | $7.3320 \times 10^{-6}$ | $1.2251 \times 10^{-5}$ | $4.1759 \times 10^{-5}$ | $0.0005$ | $8.5889 \times 10^{-6}$ | $9.0171 \times 10^{-6}$ |
| *Validation loss* | $1.0543 \times 10^{-5}$ | $1.7174 \times 10^{-5}$ | $5.9174 \times 10^{-5}$ | $0.0007$ | $1.2354 \times 10^{-5}$ | $1.2686 \times 10^{-5}$ |

During the model training process, the decrease in training loss is an important indicator for monitoring progress. However, to evaluate the model's performance, it is necessary to examine its generalizability through the validation loss and check for overfitting issues. Through heat map analysis, the correlation between level-set values and input features was examined, and the theoretical influence on training was experimentally verified (see Figure 3). Although the experiments were set for 100,000 epochs, the graph shows the range of 8000 to 10,000 epochs, making the visual differences quite apparent (see Figure 5). Table 2 presents the final loss values obtained by preventing an increase in loss through early stopping. Therefore, the loss values in Table 2 can have different epoch numbers.

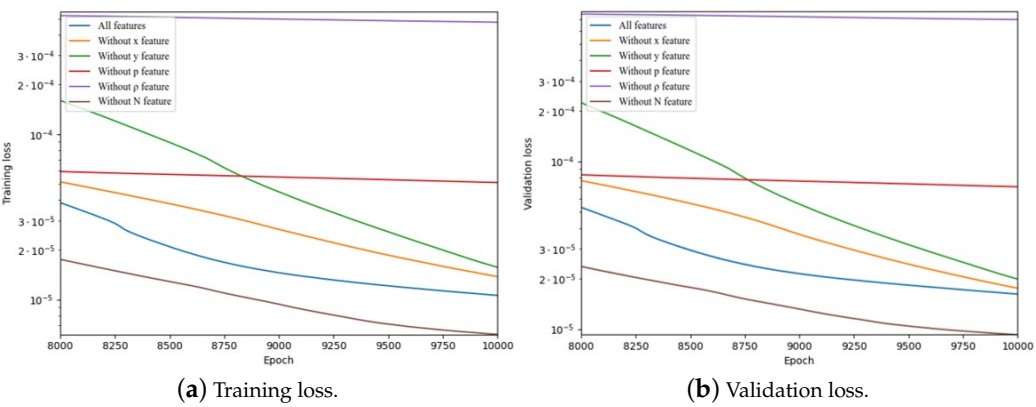

(**a**) Training loss.  (**b**) Validation loss.

**Figure 5.** Graphs of training and validation loss for the experiments based on Table 2.

From the graph concerning validation and training loss, we can confirm that among all input features, density is the most critical factor for inferring fluid isolines (see purple line in Figure 5). Density varies with the distribution of fluid particles, and to represent fluid isolines, it is essential to determine the interior/exterior of the fluid based on the level set. Density serves as a key metric for dividing these criteria, and consequently, during the training process, the loss for density was the greatest. Compared to other input features, the loss variation for density showed a significant difference, which is clearly illustrated in the chart (see Figure 5).

*Experimental Environment*

The SPH fluid simulation environment was implemented using Python version 3.9.7, and for training, Pytorch version 1.11.0 and CUDA were utilized. The computer environment used was Windows 11, AMD CPU 5800X, NVIDIA GPU RTX 3080 with 12 GB VRAM, and 64 GB RAM.

The dataset comprised 5,672,400 particles (3912 frames × 1450 particles), with 70% used for training and 30% for validation. Four models were set as comparison groups: LR, LSTM, learning representation (1-layer), and our method, where LR is the linear regression model from Lazypredict.

The learning representation (1-layer) was trained with a learning rate of 0.0001, $\frac{1}{2}$ after 50,000 epochs, and applied early stopping at 100,000 epochs. The optimizer used was Adam, with the layer configuration being linear with 5 inputs, 32 in the hidden layer, and 1 output. The loss function used was MSE, and the activation function was LeakyReLU to include negative values in the learning process.

The LSTM was trained with a learning rate of 0.0001 and applied early stopping at 500 epochs. The optimizer used was Adam, with the layer configuration set to 6 inputs for LSTM, 1000 in the hidden layers, and a linear layer with 1000 inputs, with the output set to 1. The loss function used was MSE, and the number of input frames was set to 3.

The method proposed in this paper was trained with a learning rate of 0.0001, $\frac{1}{2}$ after 50,000 epochs, and applied early stopping at 100,000 epochs. The optimizer used was Adam, with the layer configuration being linear with 5 inputs, 64 and 32 in the hidden layers, and the output set to 1. The activation function was LeakyReLU, used to include negative values in the learning process. Dropout was not used in any of the models.

Experiments were conducted in a total of five scenes for the comparison of each model and to verify the performance of the proposed model. Information will be explained in detail in the following section.

## 5. Results and Analysis

Figure 6 shows the visualization after reconstructing the surface based on the level-set inference results using linear regression and LSTM models. Figure 6a illustrates that the surface reconstruction experienced artifacts due to the divergence of the level-set values predicted via linear regression. This divergence occurs because linear regression, typically used to learn linear structures of data, fails to learn the nonlinear data structure of the level set. Figure 6a shows the first frame of the simulation, and in subsequent frames, the predicted level-set values diverge, resulting in an inaccurate representation of the fluid surfaces (see Figure 6b).

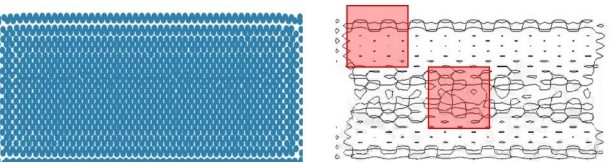

(**a**) Ground truth (left, input data), visualization of the surface predicted via linear regression (right).

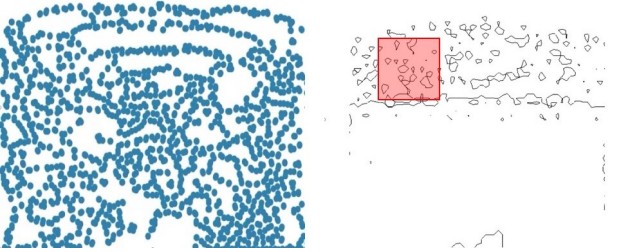

(**b**) Ground truth (left, input data), visualization of the surface predicted via LSTM (right).

**Figure 6.** Visualization of the results from predicting fluid surfaces using (**a**) linear regression and (**b**) LSTM models, respectively (red box: artifacts).

Figure 6b shows the visualization of the surface reconstruction based on the inference results from the LSTM model. Similar to linear regression, the figure shows the divergence

of the predicted level-set values using LSTM. Unlike linear regression, it is visually apparent that the level set of the initial frames can somewhat predict the surface compared to later frames. The structural characteristic of LSTM, which ensures temporal continuity by using the level-set values of previous frames as input, leads to the accumulation of errors in the predicted level-set values. This accumulation results in an increasing error as the simulation progresses, which can be visually confirmed.

Figure 7 shows a chart that calculates error values through RMSE for the level-set values inferred via the LSTM model for each frame. As previously mentioned, the error is small for the initial frames, but as the simulation progresses, the error values gradually increase, indicating a trend of growing discrepancies.

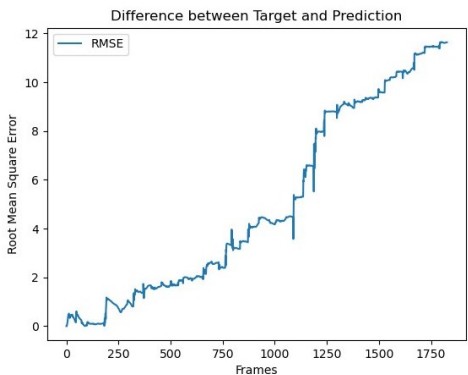

**Figure 7.** RMSE graph of the LSTM model.

Figure 8a shows the results of surface reconstruction using level-set values inferred via the learning representation (1-layer) model. Contrary to other results depicted in Figure 6, it can be visually confirmed that the surface has been somewhat tracked. However, there are artifacts where parts that should be connected are disjointed, or areas inside the fluid are incorrectly represented as the surface. Additionally, due to the lack of temporal continuity, flickering occurred when external forces not included in the training were applied during the simulation. The term "flickering" refers to the phenomenon where significant discrepancies in the reconstructed surfaces between consecutive frames lead to visually noticeable structural inconsistencies.

Figure 8b shows the surface reconstruction using level-set values inferred from the model proposed in this paper. Visually improved results can be seen compared to the learning representation (1-layer) model shown in Figure 8a, but errors still exist in the surface reconstruction. It was also confirmed that the flickering phenomenon, which appeared in Figure 8a, occurs in the proposed model as well.

This issue arises when the range of level-set values corresponding to the surface is lower than the allowed threshold. In traditional algorithms, surfaces above a certain threshold around a particle are directly represented during surface reconstruction. However, since the level-set values inferred from our method slightly differ from the ground truth, it is necessary to adjust the surface restoration using a threshold within an acceptable range. To address this issue, values had to be determined experimentally, and in this paper, 0.1 was added to the inferred $\phi$ value. As a result, the problem was resolved, and issues where the interior of the fluid was mistakenly represented as the surface, as shown in Figure 9, were corrected, allowing for the creation of smooth surfaces.

Figure 10 illustrates the differences in underestimation and overestimation of fluid isolines. In Figure 10a, the surface inferred via the 1-layer model tends to predict the surface isoline above the actual particle positions when there is significant particle movement. Additionally, the surfaces represented inside the fluid are often inferred below the particle positions, or sometimes surfaces disappear despite the presence of particles (see Figure 10a). On the other hand, the surface inferred via our method does not exhibit the problems seen with the 1-layer model's inferred surface and accurately represents the fluid's isolines.

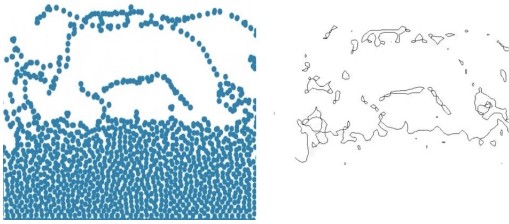

(**a**) Ground truth (left, input data), visualization of the surface predicted via the learning representation (1-layer)(right).

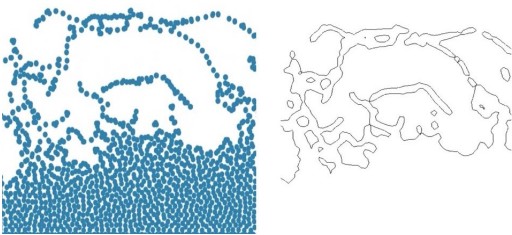

(**b**) Ground truth (left, input data), visualization of the surface predicted via our method (right).

**Figure 8.** Visualization of the results from predicting fluid surfaces using (**a**) learning representation (1-layer) and (**b**) our method, respectively.

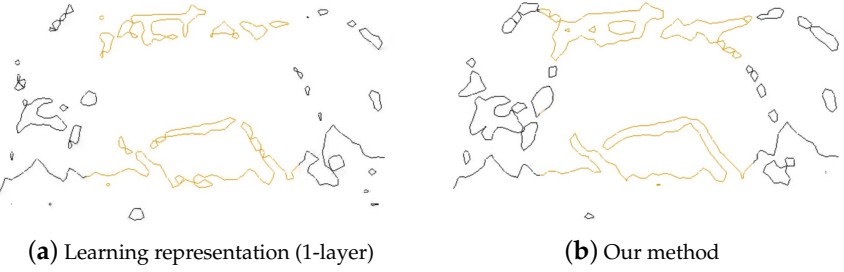

(**a**) Learning representation (1-layer)  (**b**) Our method

**Figure 9.** Comparison-1 of visual differences between the learning representation (1-layer) and our method.

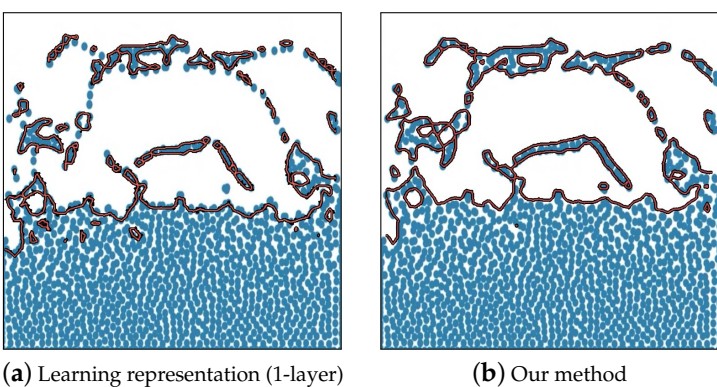

(**a**) Learning representation (1-layer)  (**b**) Our method

**Figure 10.** Comparison-2 of visual differences between the learning representation (1-layer) and our method.

Figure 11 compares the surfaces inferred by the 1-layer model and our method. As seen in Figure 11a, the surface inferred via the 1-layer model restores the surface as if it were disconnected despite the particles being connected, and even biases the position of that surface upwards. In contrast, the surface inferred using our method is accurately and stably reconstructed near where the particles exist, without bias in any direction. Similarly,

in Figure 11b, the 1-layer model restores the surface of connected particles as if they were disconnected, showing a result biased downwards. On the other hand, our method stably restores the surface at the particle positions.

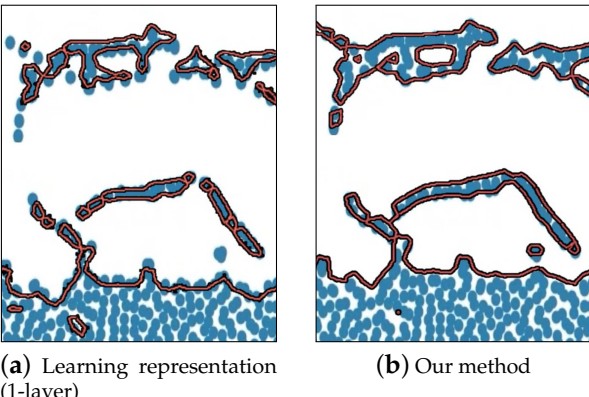

(**a**) Learning representation (1-layer)   (**b**) Our method

**Figure 11.** Comparison-3 of visual differences between the learning representation (1-layer) and our method.

Figures 12–17 show scenarios used to assess whether the trained model can be generalized and applied in various environments different from the training environment.

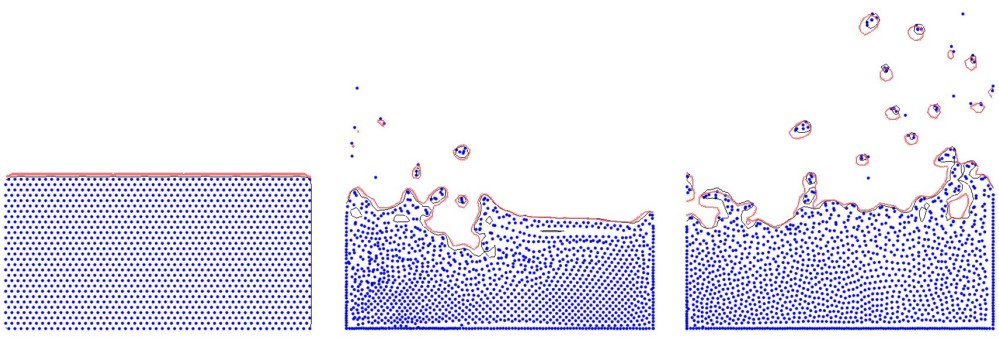

**Figure 12.** Test scene (scene name: flat surface; red line: our method; black line: ground truth).

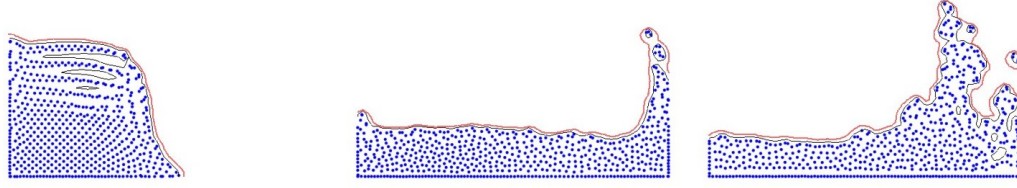

**Figure 13.** Test scene (scene name: dam break; red line: our method; black line: ground truth).

Figure 12 shows the results of extracting the isolines of fluid fluctuating due to mouse interaction on flat fluid surfaces. As depicted in the figure, there was not a significant difference in quality between the original surfaces and the isolines tracked via our method. The proposed approach is efficient as it represents isolines through learning without going through the explicit level-set calculation process, and it not only accurately represents flat water but also stably captures the shapes of bubbles exhibited in splashing effects.

Figure 13 depicts a dam breaking scene, where the fluid's isolines moving to the right due to the wave crest are well represented, showing no significant difference when compared with the original surfaces. Figure 14 presents a falling water scene, where the isolines in the splash caused by the strong impact due to gravity are accurately captured without

any omissions. While the learning representation (1-layer) produced somewhat noisy results with jagged surfaces, our method successfully represented the surfaces smoothly even in splash particles.

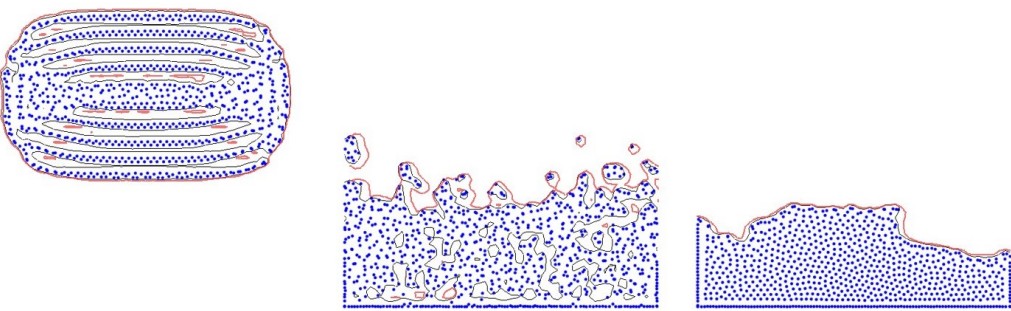

**Figure 14.** Test scene (scene name: falling water; red line: our method; black line: ground truth).

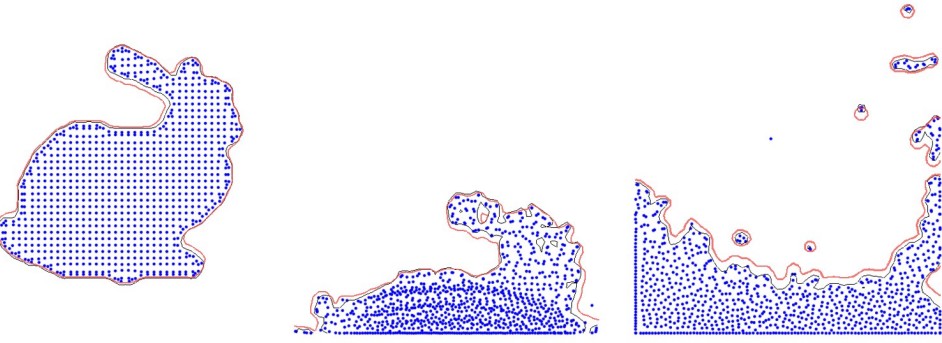

**Figure 15.** Test scene (scene name: Falling Stanford Bunny; red line: our method; black line: ground truth).

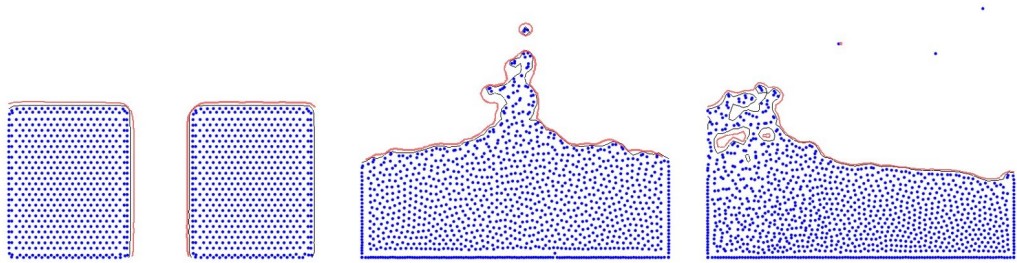

**Figure 16.** Test scene (scene name: double dam break; red line: our method; black line: ground truth).

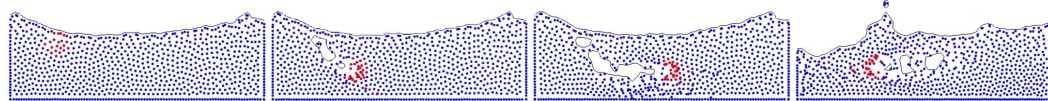

**Figure 17.** Isoline surfaces of fluid tracked in a scene that changes due to mouse interaction (red: active particle).

Figure 15 shows the results of tracking the isolines of fluid after dropping water particles in the shape of a bunny, to test on surfaces with more complex boundaries. Similar to previous results, it represented surfaces almost identical to the original surfaces and also stably represented the fluid's isolines in scattering particles, like splashes. Lastly, Figure 16 depicts a double dam breaking scene, where it accurately represented the fluid isolines in high splashes caused by the collision of two water columns and wave crests, consistent with

the results shown earlier. This paper demonstrated the restoration of complex fluid surfaces across various scenarios through experiments, consistently learning the representation of fluid surfaces in all results, not limited to specific scenes.

As indicated by the test results in Figure 18, it was observed that values with high peaks in certain sections are generated due to the influence of gravity and external forces not included in the training data. The peaks that appear between 0 and 250 frames are caused by particles falling due to gravity and then moving in the opposite direction of their initial velocity due to the rebound effect when colliding with boundaries or other particles. The remaining peaks appeared when the user dragged with the mouse, demonstrating that the method can stably track the fluid isoline even when user interaction occurs. Figure 17 shows the result of the user interfering in the fluid simulation using mouse interaction. In this scene, our method also stably tracked the surface without any flickering or missing surfaces.

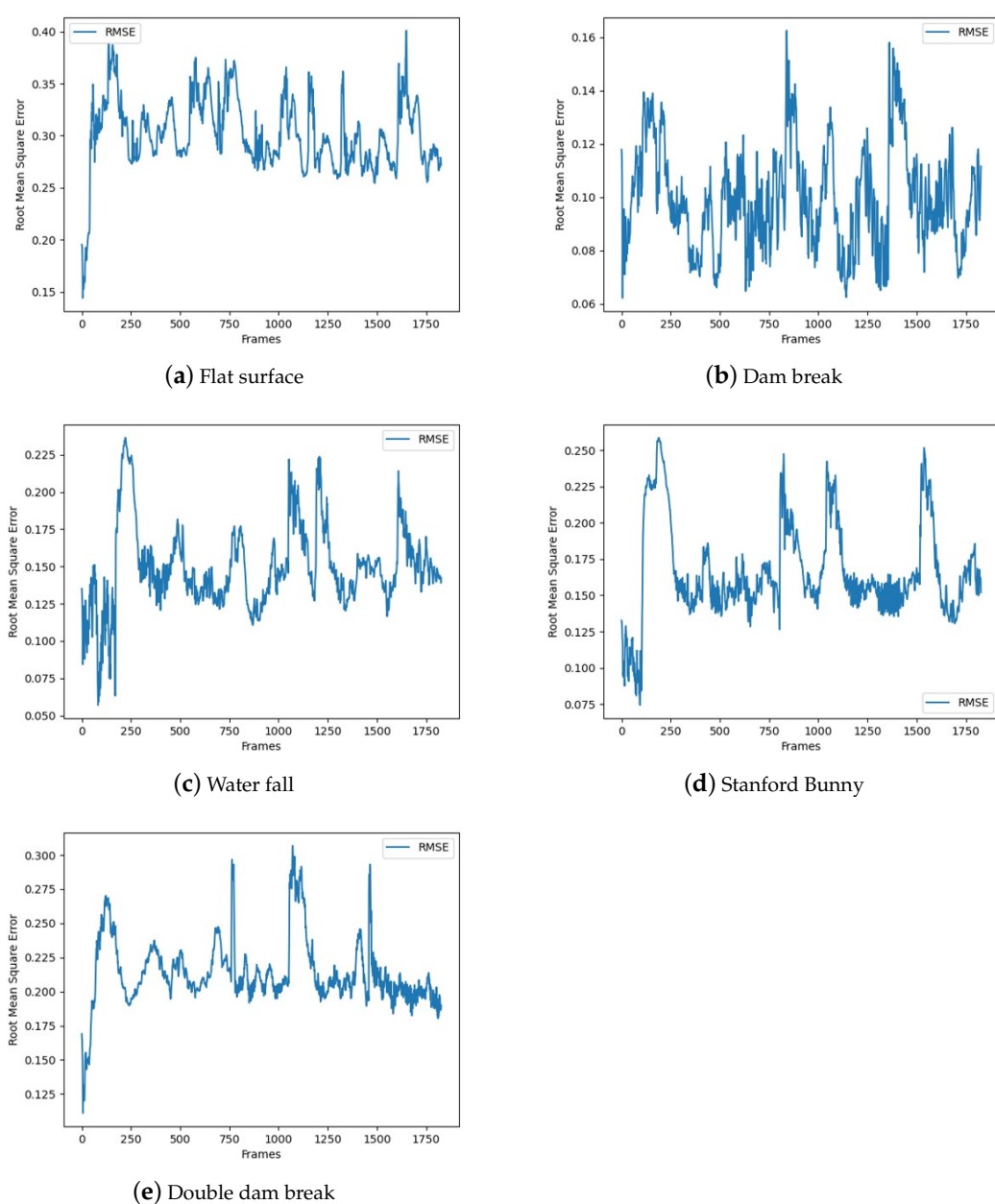

(**a**) Flat surface

(**b**) Dam break

(**c**) Water fall

(**d**) Stanford Bunny

(**e**) Double dam break

**Figure 18.** RMSE in each scene.

In particle-based simulations, a level-set function is typically required to extract surfaces or isolines. Due to the absence of connectivity information inherent in conventional

meshes, accessing neighboring particles is necessary to calculate the level set of a target particle, requiring about 15~20 neighbor particles for stable surface computation. As the total number of simulation particles increases, so does the computational load for this process. Acceleration data structures such as hash tables, K-d trees, and quadtrees are needed to solve the Nearest Neighbor Particle (NNP) problem. Moreover, kernel-based various operators are required to compute the level set from neighbor particles, making the extraction of fluid surfaces from scenes with many particles a computationally intensive task. The method proposed in this paper eliminates the need for acceleration data structures and the process of calculating level set, relying solely on the testing phase with weights learned through neural networks. This approach demands fewer resources and guarantees real-time performance in representing fluid isolines. So it can be efficiently utilized in resource-constrained environments like middleware and devices based on Android/iOS.

## 6. Conclusions and Future Research

In this paper, we proposed a learning model designed to track the isolines of a 2D fluid based on its physical properties as inputs. Compared models such as linear regression and LSTM resulted in unnatural surfaces due to the divergence of inferred level-set values. While the surface represented by the learning representation (1-layer) model was visually significant, it was not as smooth as the surface produced by the proposed model. The LSTM model, designed to ensure temporal continuity between consecutive frames, suffered from the structural characteristic of LSTM, where previous errors continuously accumulate, leading to divergence in later frames. Experiments with linear regression and learning representation (1-layer) models were conducted to see if they could reflect the data trained with a simpler structure compared to our method, but they were less accurate than the proposed model. Training with networks having a more complex structure (3-layer, Hidden layer 256, 128, 64, etc.) than our method was also attempted, but it resulted in increased training and validation loss without convergence. To verify the generality of our method, tests were conducted in a total of five scenes, and in all test scenes, visually natural fluid isolines were successfully represented.

The model proposed in this paper is independent of time, thus preventing the accumulation of errors in the level set, enabling surface reconstruction with uniform quality. Additionally, by eliminating the computational process of traditional algorithms, it saves time and resources, making it suitable for real-time isoline tracking and use in middleware.

In the future, we plan to reduce training time and improve the accuracy of the model through correlation analysis with physical properties in anisotropic forms, in addition to the input features used in this paper. Furthermore, we plan to extend our research to efficiently learn level-set inference and surface tension using models such as GNN and CNN, which were not utilized in this paper.

**Author Contributions:** Conceptualization, J.Y.K. and J.-H.K.; methodology, C.G.S.; software, J.L.; validation, J.W.L., S.-J.K. All authors have read and agreed to the published version of the manuscript.

**Funding:** This work was supported in part by the National Research Foundation of Korea (NRF) grant funded by Basic Science Research Program through the NRF funded by the Ministry of Education under Grant 2022R1F1A1063180 (Contribution Rate: 40%). This work was supported by INHA UNIVERSITY Research Grant (Contribution Rate: 20%). This research was supported by the National Research Foundation of Korea (NRF) grant funded by the Korea government (MSIT) (No. RS-2023-00254695) (Contribution Rate: 40%).

**Institutional Review Board Statement:** Not applicable.

**Informed Consent Statement:** Not applicable.

**Data Availability Statement:** The raw data supporting the conclusions of this article will be made available by the authors on request.

**Conflicts of Interest:** The authors declare no conflict of interest.

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
