# Peer review of "Isoline Tracking in Particle-Based Fluids Using Level-Set Learning Representation"

_applsci, doi:10.3390/app14062644_

Round 1
Reviewer 1 Report
Comments and Suggestions for Authors
Authors propose a novel learning model to track the isolines of fluid. The method is based on physical properties of particles analyzing the significance of weights in surface tracking. The paper aims the scope of Applied Science Journal. The work is interesting, the references are sufficient. It should be reorganized for a better understanding. Figures and table should not be placed in the middle of the page, but on top or bottom in order to make the reading fluid and without sudden interruptions. Feature analysis is very interesting and clear what happens in the proposed model.
Table 2 is difficult to read as proposed, it could be supported by a plot graph that visually clears what happens in the feature ablation.
In experiments it is not clear why methods are compared two-by-two using different input data. It should be better to show all methods together with same data over multiple examples.
Figures from 8 to 13 should be re-elaborated. It is better to overimpose the predicted results over the ground truth. The visualization chosen from authors is very hard to understand and does not help to fully understand the capability of the proposed methodology.
Caption of figure 15 reports MSE for each scene, but in plots the RMSE is represented.
Figure 15 is referenced before figure 14.
In paragraph 3.4 the authors stated that learning representation with 1-layer and 2-layers is shown, but all reported tests show only 1-layer, where is the 2-layers?
MSE loss is never shown, there is only the RMSE.
Due to the suggestions, experimental setup and results should be re-written.
Reviewer 2 Report
Comments and Suggestions for Authors
-
Artifacts generated in Figure 5 should be explicitly labeled in the figure to facilitate understanding. Additionally, figures are typically referenced after their mention in the text.
-
The paper mentions that "As the scale of the simulation increases, the time and resources consumed can be significantly reduced compared to traditional approaches." However, this statement does not seem to be substantiated in the experiments.
-
The paper utilizes learning representations for fluid boundary tracking, but the justification for choosing this approach is not fully convincing. For instance, the currently popular diffusion models appear to better represent the diffusion process of fluid particles.
- No recent references
Round 2
Reviewer 1 Report
Comments and Suggestions for Authors
The paper is more clear on many points.
It is not completely clear why the author used RMSE, are the training made on normalized output values?
Tables and figures should not be shown in the center of the page but on top or bottom. Placing them after the related text make the readability and the reading flow very difficult.
The author added figure 10 and 11 with overimposed comparison of surface prediction, but leaved the other figures without. Many figures are leaved with the same problem (i.e. figure 8, figure 12, figure 13, ...).
Figures 12 to 18 are placed among the references. This figure should be placed before of in an appendix with detailed explanation, in order to avoid reading problems.
